# Theoretical efficiency limits and speed-efficiency trade-off in myosin motors

**Andrej Vilfan**[1,2]*, **Andreja Šarlah**[3]

**1** Max Planck Institute for Dynamics and Self-Organization (MPIDS), Göttingen, Germany, **2** J. Stefan Institute, Ljubljana, Slovenia, **3** Faculty of Mathematics and Physics, University of Ljubljana, Ljubljana, Slovenia

* andrej.vilfan@ijs.si

**Data Availability Statement:** All relevant data are within the manuscript and its Supporting information files. The source code of the optimization programs is provided in the Supporting Information files (S1 Source Files).

## Abstract

Muscle myosin is a non-processive molecular motor that generates mechanical work when cooperating in large ensembles. During its cyle, each individual motor keeps attaching and detaching from the actin filament. The random nature of attachment and detachment inevitably leads to losses and imposes theoretical limits on the energetic efficiency. Here, we numerically determine the theoretical efficiency limit of a classical myosin model with a given number of mechano-chemical states. All parameters that are not bounded by physical limits (like rate limiting steps) are determined by numerical efficiency optimization. We show that the efficiency is limited by the number of states, the stiffness and the rate-limiting kinetic steps. There is a trade-off between speed and efficiency. Slow motors are optimal when most of the available free energy is allocated to the working stroke and the stiffness of their elastic element is high. Fast motors, on the other hand, work better with a lower and asymmetric stiffness and allocate a larger fraction of free energy to the release of ADP. Overall, many features found in myosins coincide with the findings from the model optimization: there are at least 3 bound states, the largest part of the working stroke takes place during the first transition, the ADP affinity is adapted differently in slow and fast myosins and there is an asymmetry in elastic elements.

## Author summary

Muscle myosin is a non-processive motor protein that can only produce sustained force when many motors are pulling on a filament collectively. During muscle contraction, myosins can achieve a high energetic efficiency exceeding 50%. Here, we discuss the fundamental physical limits to the energetic efficiency of a non-processive motor protein. We therefore reverse the question how myosin works and instead ask how a hypothetical motors with a chemical cycle like myosin's would need to work in order to achieve the maximum efficiency. The optimization result reveals many similarities with the actual myosin motors. An efficient cycle needs at least 3 bound states, and there is a trade-off between speed and efficiency, where faster motors need a lower affinity for ADP. In addition, fast motors benefit from asymmetric elasticity that allows them to pull strongly at the beginning of the stroke, but generate less drag at its end. Our results show that within

**Funding:** This work is supported by Slovenian Research Agency (ARRS), Grant No. P1-0099 to A. V., Grant No. P1-0060 to A.Š. and Grant No. N1-0197 to A.Š. The funders had no role in study design, data collection and analysis, decision to publish, or preparation of the manuscript.

**Competing interests:** The authors have declared that no competing interests exist.

physical limits, the working cycles of different myosin isoforms are well adapted to maximize efficiency under different conditions.

## Introduction

Motor proteins convert chemical energy, usually gained from the hydrolysis of ATP, into mechanical work [1, 2]. They power a number of essential processes in the cell including intracellular transport, cell division, muscle contraction, and beating of cilia and flagella. Many motor proteins can reach remarkably high energetic efficiencies. Muscle myosin achieves an efficiency of more than 50% [3–11]. The efficiency of the $F_1$ ATP synthase has been reported as close to 100% within experimental error [12, 13]. The hypothesis that biological systems have evolved to perform their tasks with maximum possible efficiency [14] has been proposed and tested in systems as diverse as transcriptional networks [15], fluid propulsion [16] and chemosensing [17] by cilia, arrangement of teeth [18] and many more.

The efficiency of molecular engines and the source of their losses has also been studied intensely from the perspective of stochastic thermodynamics. Most of the work has focused on processive motors, where a single motor molecule (typically dimeric) can transport a cargo by taking many consecutive steps along a filament [19–21]. There is no upper limit, other than 100%, on the efficiency of a processive motor, provided that the stepping is tightly coupled to the hydrolysis of ATP (each forward step is coupled to the hydrolysis of one ATP molecule and each backward step with the synthesis of one) and that the motor steps very slowly relative to its chemical kinetics (i.e., it is close to stall). At a finite speed, the transitions in the working cycle are out of equilibrium and linked to entropy production. In order to maximize the efficiency at a certain velocity, the largest part of the dissipation has to be allocated to the rate-limiting steps [22, 23]. There is also a relationship between the efficiency and the randomness of stepping: restricting the fraction of backward steps imposes a limit on the energetic efficiency [24, 25]. Real motor proteins, notably kinesin, however do not come close to efficiencies that would be theoretically possible from the thermodynamic uncertainty relationships [26, 27]. The additional losses typically originate from loose coupling between the chemical reaction and mechanical stepping [28]. Imperfect coordination between heads of a dimeric motor is also a common source of losses [29]. Non-processive motor proteins, such as muscle myosin, can only generate forces when cooperating in larger ensembles, because a single motor detaches from the track and reattaches as part of its working cycle [30, 31]. The attachment and detachment of a motor introduce additional randomness into the kinematics of the cycle. One can expect that the resulting losses reduce the upper limit on efficiency.

Physiologically, the relevant efficiency measure depends on the function of the motor. For motors that need to maintain a high power density (rate of work per unit volume of muscle), efficiency at a given power per motor will be decisive, because the volume density of motors is largely conserved. On the other hand, if a high force density (force per unit cross-section area) is needed, efficiency at a given average force per motor should be maximized. Likewise, for motors that need a high velocity, the relevant quantity is efficiency at a prescribed velocity. Alternative quantities, such as the efficiency at maximum power (EMP) have also been studied, in particular for processive motors [32–36]. Here, we concentrate on the efficiency at a given velocity, which we consider as most relevant for muscle myosin.

In this paper, we study a mechano-chemical model of a myosin motor. Our aim is to determine the efficiency limits under expected constraints and find out which properties of myosin can be understood as adaptations to maximize efficiency. In order to find the efficiency limits,

we only constrain a small number of parameters that are physically limited (like the stiffness and the nucleotide binding rates) and determine all other parameters by numerical optimization.

## Model and methods

In our description of the non-processive motor we follow T.L. Hill's formalism [37], which is the basis of most myosin models. The duty cycle consists of attaching to the track (actin filament), a conformational change in the lever (working stroke), release of hydrolysis products and binding of a new ATP molecule followed by detachment, reverse conformational change (recovery stroke) and ATP hydrolysis. We describe the cycle of the motor with $N_S$ chemical states, of which $N_B$ are bound states (Fig 1A). Each state represents a unique conformation, characterized by the unstrained lever position $d_i$ and the free energy $G_i$. With $X$ we denote the position of the track relative to the backbone (thick filament), with $x_A$ the position of the binding site on the track and with $x_M$ the unstrained position of a motor on the backbone (Fig 1B). The strain on a motor is $X + x_A - (x_M + d_i) = x - d_i$ and its elastic energy $U_i(x) = U(x - d_i)$. In the simplest case of a harmonic potential, $U(x) = Kx^2/2$ with a spring constant $K$ (Fig 1C). The transitions are stochastic with rates

$$k_i(x) = k_i^0 \exp(-\alpha_i(U_{i+1}(x) - U_i(x))/(k_B T)). \tag{1}$$

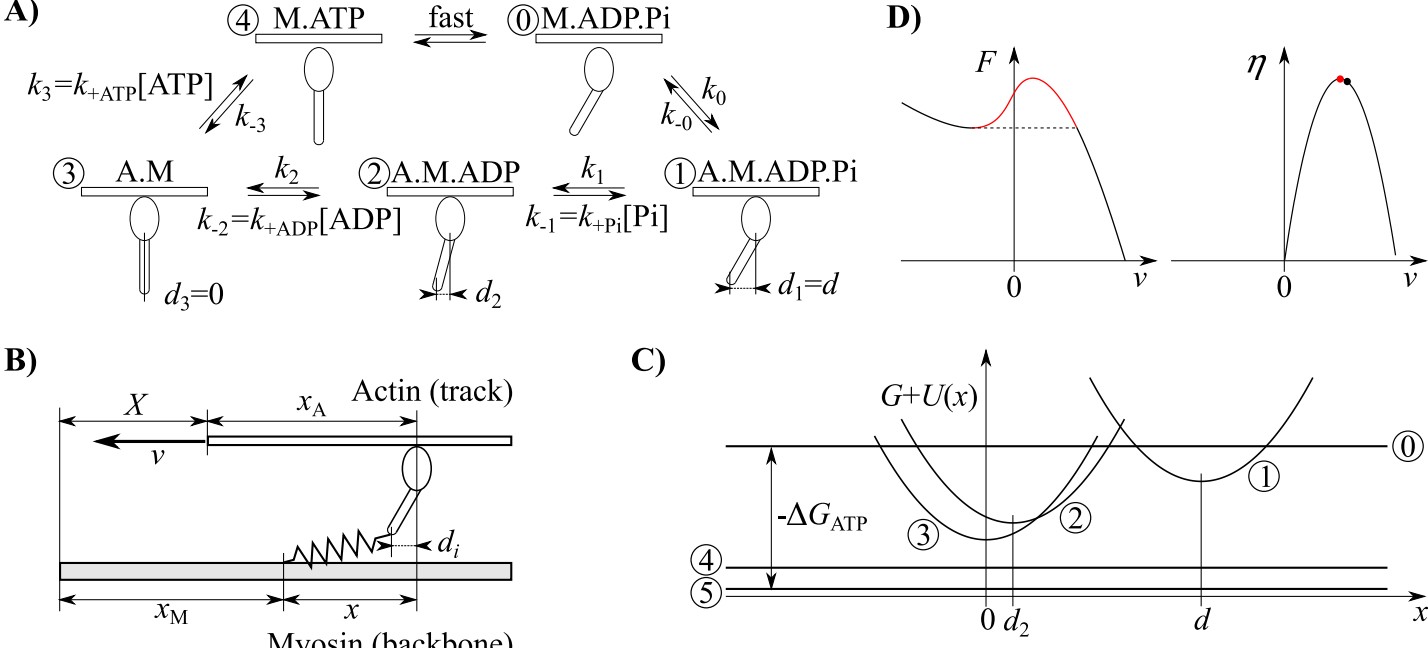

**Fig 1. Model definition.** (A) A kinetic scheme for myosin with $N_B = 3$ bound and 2 unbound states. A myosin head with ADP and Pi binds to actin in state 1. After releasing Pi it enters state 2 and subsequently state 3 (rigor) after releasing ADP. Upon binding a new ATP molecule, it detaches from actin (4) and hydrolyzes ATP (0). (B) The elastic model: the actin and myosin filaments slide past each other with velocity $v$. A myosin head is connected to its filament through an elastic element with strain $x - d_i$. (C) Free energy landscape: the state $i$ has free energy $G_i$ if detached and includes the elastic energy, $G_i + U_i(x)$, if attached. (D) For certain parameters, the force-velocity relationship (left diagram) can contain an anomalous region with a positive slope (red). If the motors are working in a fixed-force (as opposed to fixed velocity) configuration, these solutions are unstable and we only consider the maximum efficiency (right diagram) for operational points outside the hysteresis (black dot).

The reverse transitions have the rates

$$k_{-i}(x) = k^0_{-i} \exp((1 - \alpha_i)(U_{i+1}(x) - U_i(x))/(k_B T)), \tag{2}$$

where $k^0_{-i}/k^0_i = \exp((G_{i+1} - G_i)/(k_B T))$. For transitions that involve attachment (state $i$ unbound and state $i + 1$ bound) we set $\alpha_i = 1$, meaning that the strain only affects the attachment rate, while the detachment rate is constant. The strain dependence of the attachment rate is indirect and reflects the fact that in order to reach a binding site, the elastic element has to stretch through thermal fluctuations. Because of the distances involved, we consider this dependence stronger than the strain dependence of the detachment rate and neglect the latter. Likewise, we set $\alpha_i = 0$ for detachment transitions. If a transition involves binding of ATP, ADP, or Pi, the corresponding rate $k^0$ is $k_{+\text{ATP}}[\text{ATP}]$, $k_{+\text{ADP}}[\text{ADP}]$ or $k_{+\text{Pi}}[\text{Pi}]$, respectively.

For each state $i$ we define $P_i(x)$ as the probability density to find the motor in that state with strain $x$. The probabilities are normalized to $\sum_{i=1}^{N_S} \int_{-\infty}^{\infty} P_i(x) dx = 1$. When the filaments move with velocity $v = -dX/dt$ against each other, the strain on the motors changes according to $dx/dt = -v$. The system is described by the following set of master equations

$$\left(\frac{\partial}{\partial t} - v\frac{\partial}{\partial x}\right) P_i(x, t) = j_{i-1}(x, t) - j_i(x, t) \tag{3}$$

with the probability flux densities

$$j_i(x, t) = k_i(x)P_i(x, t) - k_{-i}(x)P_{i+1}(x, t) . \tag{4}$$

In detached states, the positions are quickly thermally equilibrated,

$$P_i(x, t) = P_i^T(t)\frac{e^{-U_i(x)/(k_B T)}}{\int_{-\infty}^{\infty} e^{-U_i(x')/(k_B T)} dx'} \tag{5}$$

and the corresponding master equations are

$$\frac{\partial}{\partial t} P_i^T(t) = \int_{-\infty}^{\infty} [j_{i-1}(x, t) - j_i(x, t)] dx . \tag{6}$$

State $i + N_S$ is identical to state $i$, but because one ATP molecule has been hydrolyzed during the cycle, its free energy is $G_{i+N_S} = G_i + \Delta G_{\text{ATP}}$. For a constant velocity $v$, the probability densities are determined by the stationary solution of Eqs (3) and (6). Then the total flux is conserved around the cycle and corresponds to the total rate of ATP hydrolysis per motor,

$$\int_{-\infty}^{\infty} j_i(x) dx = r_{\text{ATPase}} . \tag{7}$$

The average force per motor follows as

$$F = \sum_{i=1}^{N_B} \int_{-\infty}^{\infty} P_i(x)U_i'(x) dx \tag{8}$$

where $U_i' = \partial U_i/\partial x$ is the spatial derivative.

To understand the origin of dissipation in the working cycle of myosin, we can compute the entropy production rates for individual transitions. At position $x$, each state can be assigned a chemical potential

$$\mu_i(x) = G_i + U_i(x) + k_B T \ln P_i(x) . \tag{9}$$

The transitions are generally out of equilibrium and lead to entropy production with density

$$\dot{s} = j_i(\mu_i - \mu_{i+1})/T \ . \tag{10}$$

The total dissipation rate is obtained by summation over all transitions and integration over all positions $x$. One can verify that the total dissipation (time derivative of the total entropy in the system and the heat bath) equals the free energy of ATP hydrolysis per unit time, reduced by the power output [38]:

$$
\begin{aligned}
T\dot{S} &= \sum_{i=1}^{N_S} \int_{-\infty}^{\infty} j_i[\mu_i - \mu_{i+1}]dx \\
&= -\int_{-\infty}^{\infty} \left[ \sum_{i=1}^{N_S} (j_{i-1} - j_i)\mu_i + j_{N_S}(\mu_1 - \mu_{N_S+1}) \right] dx \\
&= \int_{-\infty}^{\infty} \left[ \sum_{i=1}^{N_S} v\frac{\partial P_i}{\partial x}(G_i + U_i + k_B T \ln P_i) - j_{N_S}\Delta G_{ATP} \right] dx \\
&= -vF + r_{ATPase}(-\Delta G_{ATP}) \ .
\end{aligned}
\tag{11}
$$

In the second line we re-ordered the terms in the sum, in the third we applied Eqs (3) and (9) and in the last we performed partial integration. The efficiency follows as

$$\eta = 1 - \frac{T\dot{S}}{r_{ATPase}(-\Delta G_{ATP})} \ . \tag{12}$$

The model equations can be non-dimensionalized by expressing all energies with the thermal energy $k_B T$ and all distances with the power stroke size $d$. The velocity is expressed with $k_{+ATP}^{max}[ATP]d$, where $k_{+ATP}^{max}[ATP]$ is the rate of ATP binding. While most model parameters are obtained by numerical optimization, there are three dimensionless parameters that are physically restricted. First is the dimensionless free energy of ATP hydrolysis $-\Delta G_{ATP}/k_B T$, which is close to 25 under physiological conditions [5]. The second is the dimensionless stiffness which is constrained because of the elastic nature of a protein structure. We express it as the elastic energy of the spring when displaced by the working stroke distance $d$, in units of thermal energy: $\frac{1}{2}Kd^2/k_B T$. For myosin we estimate $K = 3\,\text{pN/nm}$ and $d = 8\,\text{nm}$ [39], which gives $\frac{1}{2}Kd^2/k_B T = 24$. In addition, in models with three bound states ($N_B = 3$) and a prescribed velocity, the binding rate of ADP is also a fixed parameter and expressed with the dimensionless rate $k_{+ADP}[ADP]/(k_{+ATP}[ATP])$.

Furthermore, the efficiency is independent of the motors' duty ratio (fraction of time it spends in the bound state), because the dwell time in the detached state slows down the cycle and proportionally reduces the force and the ATP consumption, but does not affect their ratio. Therefore, the relative free energy level of the detached state, $G_0 - G_1$ does not appear as a relevant optimization parameter. In cases where we only discuss the maximum efficiency, independent of velocity, the dimensionless velocity also becomes an optimization parameter.

## Numerical solution

For a given set of parameters we determined the stationary solution ($\partial P_i/\partial t = 0$) of the system of coupled linear differential equations given by Eq (3) with the boundary condition $P_i(x \to \infty) = 0$ for $i = 1\ldots N_B$. We carried out the numerical integration in negative direction in $x$ using a non-adaptive 3-step backward differentiation formula (BDF) method, which is

**Table 1. List of parameters.** The parameters are either fixed or subject to optimization (OPT) without or with constraints (C).

| Parameter | Fig 2A | | Fig 2B | | | Fig 3A |
|---|---|---|---|---|---|---|
| $\nu/(k_{+\mathrm{ATP}}^{\max}[\mathrm{ATP}]d)$ | OPT | | OPT | | | parameter |
| $-\Delta G_{\mathrm{ATP}}/k_B T$ | parameter | | 25 | | | 25 |
| $Kd^2/(2k_B T)$ | OPT, C | | OPT, constrained by parameter | | | OPT, C[1] |
| $N_B$ | 2 | 3 | 2 | 3 | 4 | 3 |
| $G_0/k_B T$ | 0 (by definition) | | | | | |
| $G_1/k_B T$ | 0 (no effect on efficiency) | | | | | |
| $G_2/k_B T$ | OPT | OPT | OPT | OPT | OPT | OPT |
| $G_3/k_B T$ | $\Delta G_{\mathrm{ATP}}/k_B T$ | OPT | $\Delta G_{\mathrm{ATP}}/k_B T$ | OPT | OPT | OPT |
| $G_4/k_B T$ | | $\Delta G_{\mathrm{ATP}}/k_B T$ | | $\Delta G_{\mathrm{ATP}}/k_B T$ | OPT | $\Delta G_{\mathrm{ATP}}/k_B T$ |
| $G_5/k_B T$ | | | | | $\Delta G_{\mathrm{ATP}}/k_B T$ | |
| $\alpha_0$ | 1 | 1 | 1 | 1 | 1 | 1 |
| $\alpha_1$ | OPT | OPT | OPT | OPT | OPT | OPT |
| $\alpha_2$ | 0 | OPT | 0 | OPT | OPT | OPT |
| $\alpha_3$ | | 0 | | 0 | OPT | 0 |
| $\alpha_4$ | | | | | 0 | |
| $k_0^0/k_{+\mathrm{ATP}}^{\max}[\mathrm{ATP}]$ | OPT | OPT | OPT | OPT | OPT | OPT |
| $k_1^0/k_{+\mathrm{ATP}}^{\max}[\mathrm{ATP}]$ | OPT | OPT | OPT | OPT | OPT | OPT |
| $k_2^0/k_{+\mathrm{ATP}}^{\max}[\mathrm{ATP}]$ | 1 | OPT | 1 | OPT | OPT | OPT, C[2] |
| $k_3^0/k_{+\mathrm{ATP}}^{\max}[\mathrm{ATP}]$ | | 1 | | 1 | OPT | OPT, C |
| $k_4^0/k_{+\mathrm{ATP}}^{\max}[\mathrm{ATP}]$ | | | | | 1 | |
| $d_1/d$ | 1 (by definition) | | | | | |
| $d_2/d$ | 0 | OPT | 0 | OPT | OPT | OPT |
| $d_3/d$ | | 0 | | 0 | OPT | 0 |
| $d_4/d$ | | | | | 0 | |

[1] functional optimization with a constraint on $U''$ for anharmonic springs

[2] indirectly constrained through the bound on $k_{-\mathrm{ADP}}[\mathrm{ADP}]$

suited for handling stiff ODEs. Simultaneously, we integrated the rate of ATP consumption given by Eq (7) and force by Eq (8) (see S1 Source Files). The resulting efficiency is calculated as $\eta = \nu F/(-r_{\mathrm{ATPase}}\Delta G_{\mathrm{ATP}})$. The efficiency as a function of the parameters that are listed in Table 1 for each case is used by a numerical optimizer using a quasi-Newton algorithm (routine E04JYF from the NAG Library, Numerical Algorithms Group). Solutions that only considered operating points outside a hysteresis in the force-velocity relation were obtained using a sequential quadratic programming procedure with nonlinear constraints (NAG routine E04UCF). Parameter scans were made by initializing the optimizer with the previous solution, once in ascending and once in descending direction, and additional verifications with different initial parameter values were carried out. The values of optimization parameters for all solutions shown in this paper are given in S1 Data.

## Validity of the stationary solution

Our numerical solution assumes that the motor ensemble moves at a constant velocity during contraction. In myosin, this stationarity has often been questioned. In fact, some studies show that under high loads (close to stall) myosin motors can synchronize their cycles, leading to a step-wise contraction [40–42]. Signatures of such steps can be seen in muscle transiently after

changes of load [43]. Kaya *et al.* reported step-wise motion in small groups of myosin motors [44] under high load. In these studies, the coordinated stepping appears in a narrow regime close to isometric conditions. We therefore do not expect it to affect the maximum efficiency, which is achieved at a lower load and higher velocity. Furthermore, recent direct observations with high-speed AFM (atomic force microscopy) showed no cooperativity between myosins on non-regulated thin filaments [45].

We verified the validity of the stationary solution by comparing our numerical solutions with stochastic simulations on a finite ensemble of motors. For a selection of optimal parameter sets, we simulated a group of $N_m$ motors pulling against a constant load. The simulations were carried out with a Gillespie algorithm and rapid mechanical equilibration at each step [42]. All results show a very good convergence for two-digit motor numbers (S1–S4 Figs). The number of motors needed is similar to the numbers that were needed to achieve a high efficiency in single-filament experiments [46]. No coordinated steps are visible in the simulated traces. The efficiencies obtained from the simulation converge towards the results of the stationary solution, and the velocity becomes constant in time for large motor numbers.

## Results and discussion

### Velocity-independent maximum efficiency

We first numerically determine the maximum efficiency $\eta^{\mathrm{max}}$ for a system with $N_B = 2, 3, \ldots$ bound states. Because the velocity is unconstrained we always evaluate the efficiency at the peak of the $\eta$-$v$ relationship (see Fig 1D for an example). If we put no constraint on the kinetic rates, $\eta$ also depends on the elastic constant $\frac{1}{2}Kd^2/k_BT$, the free energy differences between bound states $G_i/k_BT$, the ratios between kinetic constants $k_i/k_j$, coefficients $\alpha_i$ and for $N_B > 2$ on the lever position in the intermediate states $d_i$ for $2 \leq i \leq N_B - 1$. We also fixed the parameters that do not influence the optimization outcome (i.e., one kinetic constant, a constant offset in free energy levels and the free energy difference between unbound and bound steps). All other parameters have been obtained by a multidimensional optimization procedure. All parameters of the model are listed in Table 1.

The results are shown in Fig 2. The dependence of maximum efficiency on the free energy of ATP hydrolysis is non-monotonic (Fig 2A). If the ATP, ADP and Pi concentrations are close to chemical equilibrium, $-\Delta G_{\mathrm{ATP}}/k_BT \ll 1$, the efficiency is $\eta^{\mathrm{max}} = 0.162$ for $N_B = 2$. A higher efficiency is possible if the nucleotide concentrations are further away from equilibrium. A similar effect, namely an efficiency that is maximal outside the linear response regime, has already been observed in some scenarios with Brownian ratchets [47]. For $-\Delta G_{\mathrm{ATP}}/k_BT = 25$ (physiological value), the efficiency limit is 42%. However, this efficiency is only achievable if the motor ensemble is working with a prescribed velocity. Namely, the maximum efficiency is found in a region with an anomalous force-velocity relationship (Fig 1D). In a more common fixed-force scenario, the ensemble cannot be held at that point without jumping also to a less efficient point with the same force. In the fixed-force regime, we therefore impose a constraint that the operational point has to be outside the hysteresis. This reduces the maximum possible efficiency to 38% (thin solid line in Fig 2A). The origins of dissipation are analyzed in Fig 2C, which shows the rate of entropy production (Eq (10)) for each transition and each head position $x$ (equivalent of strain). The losses mainly result from a premature detachment of strained heads (red line in Fig 2C). The model with $N_B = 3$ bound states, on the other hand, can already achieve very high efficiencies, but only with very stiff springs (dotted lines in Fig 2A). Here the attachment/detachment of heads occurs primarily from/to the position close to the unstrained state (Fig 2D).

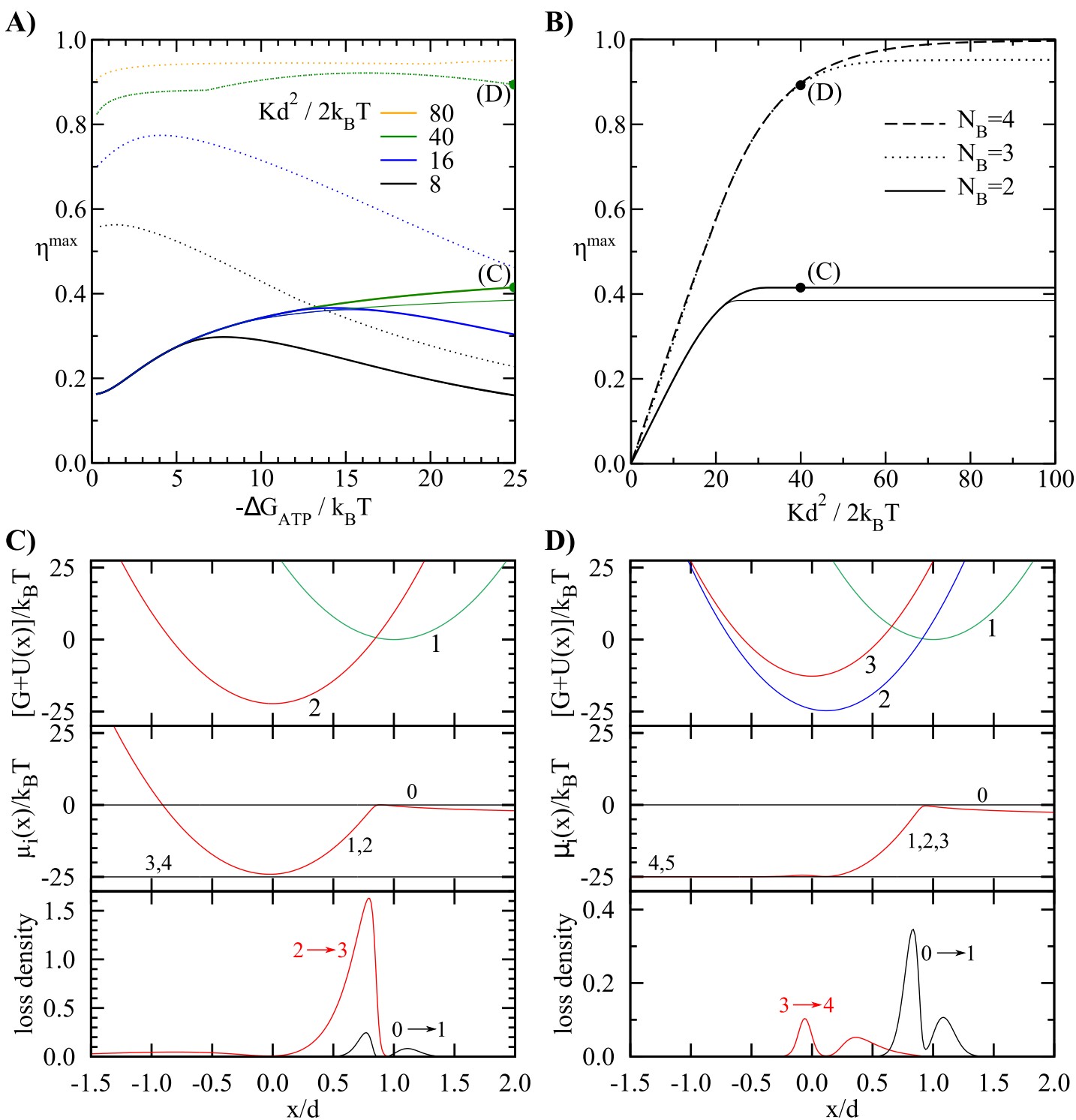

**Fig 2. Maximum efficiency without prescribed velocity.** (A) Maximum efficiency as a function of the available free energy $-\Delta G_{ATP}/k_B T$ with $N_B = 2$ (solid lines) and $N_B = 3$ (dotted lines) bound states. Different colors represent different values of the maximally allowed dimensionless elastic constant $\frac{1}{2} K d^2/k_B T$. For 2 bound states, the efficiency reaches its maximum of 42% with a moderate stiffness. Thin lines show the efficiency if only stable solutions in a fixed-force regime are considered. The system with 3 bound states can in theory achieve efficiencies well above 90%, but only with an extremely high stiffness. The two green dots mark the parameters used in panels (C) and (D). (B) Maximum efficiency with 2,3, and 4 bound states and $-\Delta G_{ATP}/k_B T = 25$ as a function of the bound on the elastic constant. The thin lines show the results for maximum fixed-force efficiency. (c,d) Potentials $G_i + U_i(x)$ with optimal parameters, chemical potentials $\mu_i(x)$, and dimensionless loss densities, defined as $j_i(\mu_i - \mu_{i+1})d/(r_{ATPase}|\Delta G_{ATP}|)$, for $N_B = 2$ (C) and $N_B = 3$ (D). Transitions between the bound states are fast and the states are in a thermal equilibrium, thus, the chemical potentials of bound states are equal. Parameters: $-\Delta G_{ATP}/k_B T = 25$, $\frac{1}{2} K d^2/k_B T = 40$ and (C) $v/(k_{+ATP}^{max}[ATP]d) = 0.406$, $G_2/k_B T = -22.3$, $k_0^0/k_{+ATP}^{max}[ATP] = 4.12$; (D) $v/(k_0^0 d) = 0.147$, $G_2/k_B T = -24.8$, $G_3/k_B T = -12.8$, $d_2/d = 0.119$.

The maximum efficiency as a function of the elastic constant $K$ is shown in Fig 2B. One expects that the elastic energy stored into the spring during the working stroke imposes an upper limit on the useful work, which gives an efficiency of $\eta \leq \frac{1}{2} K d^2 / |\Delta G_{\mathrm{ATP}}|$. In the following, we provide an analytical argument for the efficiency bound in the limit of very soft springs, $K d^2 \ll |\Delta G_{\mathrm{ATP}}|$ (S5 Fig). In this limit the forces and the elastic energies become small and lose the influence on the transition rates. The rates can be regarded as irreversible. Models with strain-independent transition rates can be solved analytically [31]. The unloaded velocity is maximal when the first transition is fast $k_1 \rightarrow \infty$ and contains the full working stroke ($d_1 = d$, $d_2 = \ldots = d_{N_B} = 0$), with the remaining transitions having equal rate constants $k_2 = \ldots = k_{N_B}$. It has the value $v^{\max} = 2d(1 - 1/N_B)/t_{\mathrm{on}}$, where $t_{\mathrm{on}} = (k_1^{-1} + \ldots + k_{N_B}^{-1})$ [48]. The maximum force per motor is $K d t_{\mathrm{on}} / t_{\mathrm{cycle}}$, while the ATPase rate is always $1/t_{\mathrm{cycle}}$. The maximum power and efficiency are achieved at 1/2 the stall force and 1/2 the maximum velocity. At that point, the work per ATP is $\frac{1}{2} K d^2 (1 - 1/N_B)$ and the efficiency is $\eta = [K d^2 / (-2\Delta G_{\mathrm{ATP}})](1 - 1/N_B)$. Therefore, the maximum efficiency in the limit of soft springs grows linearly with the stiffness with a proportionality constant that depends on $N_B$ (Fig 2B). The upper limit of $\frac{1}{2} K d^2$ can only be reached with an infinite number of bound states. This solution reveals another interesting observation, namely that the transitions between bound states are not necessarily fast in the optimal regime. Although a slow transition increases dissipation, a cascade of transitions with similar rate constants narrows the distribution of times between attachment and detachment, i.e., it makes it more deterministic (S5(D) Fig). A narrower distribution means that the elastic energy of strained springs just before detachment, which is another source of dissipation, will be smaller. In the extreme limit of fully deterministic attachment times, it is even possible that all motors detach with exactly zero strain. This result is in contrast with processive motors, where increasing the kinetic constants always improves the efficiency at a given velocity [22].

## Maximum efficiency at a given velocity

In the second scenario we determine the maximum efficiency at a given velocity $v$ while imposing upper limits on two rate constants: the rate of ATP binding with detachment $k_{+\mathrm{ATP}}^{\max}$ and the rate of ADP binding $k_{+\mathrm{ADP}}^{\max}$. Both rate constants are of a similar order of magnitude, $\sim 1\,\mu\mathrm{M}^{-1}\mathrm{s}^{-1}$, in most myosins [49, 50], but because of the lower ADP concentration [5], we expect a ratio of $k_{+\mathrm{ADP}}[\mathrm{ADP}]/k_{+\mathrm{ATP}}[\mathrm{ATP}] \sim 1/100$. The maximum efficiency for different ratios between the two limiting rates is shown in Fig 3A. It displays a trade-off between speed and efficiency. In the limit of a very slow sliding velocity the maximum efficiency with constrained kinetic rates reaches the values from the previous section, independent of the ratio between the bounds on kinetic rates. The fast transition between the first bound states contains most of the working stroke ($d_2 \ll d_1$). The optimal stiffness is at its upper limit which allows for the high exerted force and a better control on the strain dependent detachment rate. For high sliding velocities the maximum efficiency is decreased and a large part of the working stroke is allocated to the transition with the limited kinetic rate ($d_2 \sim d_1$) The optimal stiffness becomes softer to reduce negative forces caused by the imprecise position of attachment and detachment of heads.

An interesting outcome of the optimization is that the free energy difference between the last two bound states $G_{N_B-1} - G_{N_B}$ depends strongly on the velocity for which the motor is optimized. At low velocities, an uphill transition slows down the detachment and augments its strain sensitivity, see the potentials $G_i + U_i(x)$ in the top panel of Fig 2D. At the same time, a low energy of the penultimate bound state increases the energy difference available for the

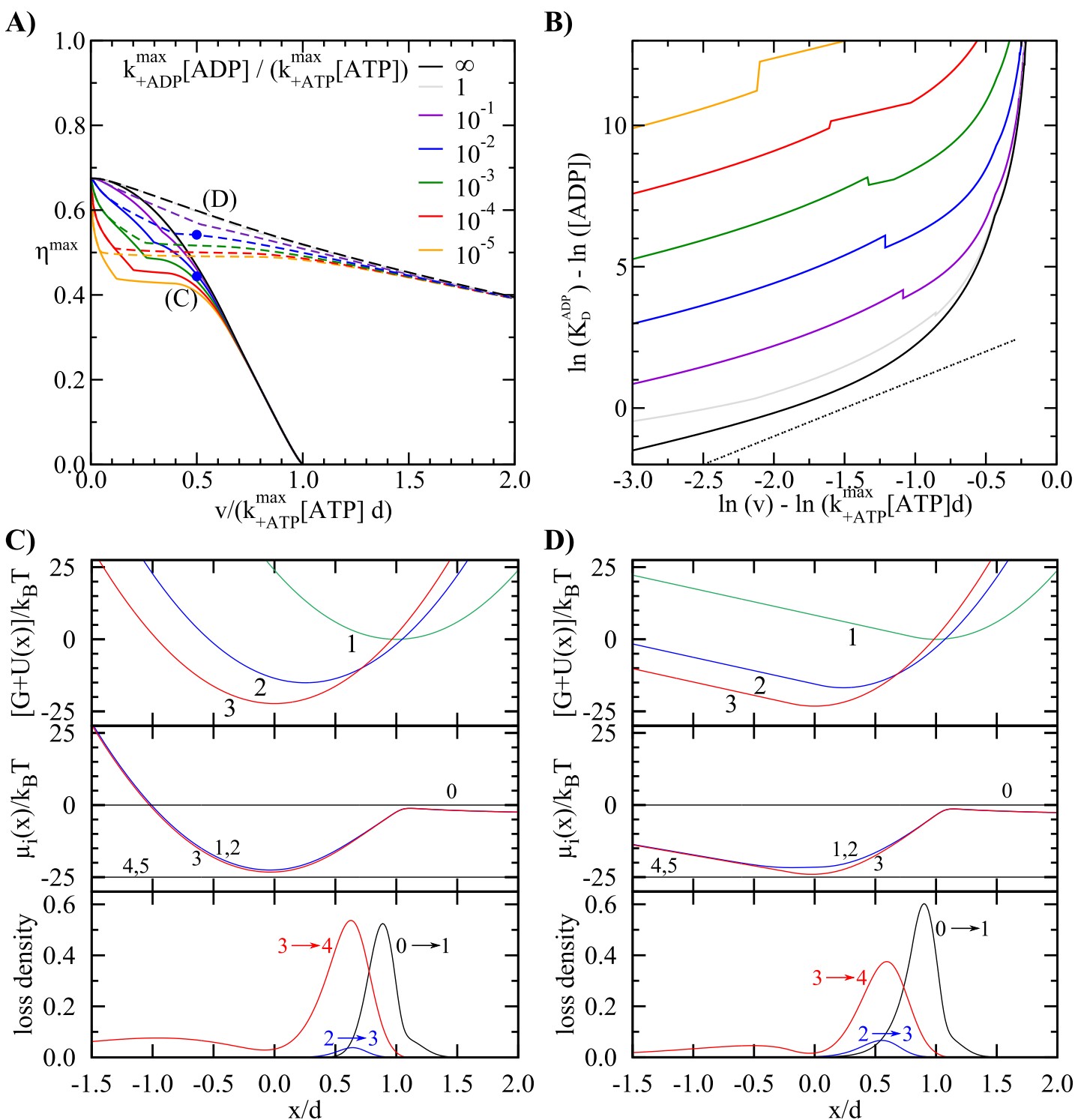

**Fig 3. Optimal efficiency with restricted kinetic rates.** (A) Maximum efficiency as a function of the sliding velocity for which the motor is optimized in a scenario with $N_B = 3$ and two rate limiting steps: the ATP-dependent detachment ($k_{+ATP}^{max}[\text{ATP}]$) and the ADP binding ($k_{+ADP}^{max}[\text{ADP}]$). The black line shows the case of fast ADP binding/unbinding. The dashed lines show the optimal efficiency when an arbitrary elastic potential $U(x)$ is allowed. In both cases the constraint $0 \leq U''(x)d^2/(2k_BT) \leq 24$ applies. The blue dots mark the parameters used in panels (C) and (D). (B) Dissociation constant for ADP, representative for the free energy difference between states $N_B$ and $N_B - 1$, as a function of the velocity for which the parameters are optimized. The dashed line shows a quadratic dependence, which has been seen experimentally when comparing myosins from different muscle types [51]. (c,d) Potentials $G_i + U_i(x)$, chemical potentials $\mu_i(x)$ and loss densities at the points marked in panel (A). Parameters: (C) $G_2/k_BT = -15.1$, $G_3/k_BT = -22.3$, $d_2/d = 0.25$, $\alpha_2 = 0$, $k_0^0/k_{+ATP}^{max}[\text{ATP}] = 2.33$, (D) $G_2/k_BT = -16.8$, $G_3/k_BT = -23.2$, $d_2/d = 0.24$, $\alpha_2 = 0$, $k_0^0/k_{+ATP}^{max}[\text{ATP}] = 1.91$, $x_a/d = -0.19$.

working stroke. On the other hand, motors optimized for high velocities have a strong down-hill transition before detachment, see the top panel of Fig 3C. In this case, quick detachment of motors is more important than the precise control over the position where the detachment takes place. In myosin the last transition between the bound states is the release of ADP. The free energy difference of this transition can be expressed with the ADP dissociation constant $K_D^{\mathrm{ADP}} = k_{-\mathrm{ADP}}/k_{+\mathrm{ADP}}$ as $G_0 - G_{\mathrm{ADP}} = k_B T \ln([\mathrm{ADP}]/K_D^{\mathrm{ADP}})$. The resulting optimal $K_D^{\mathrm{ADP}}$ as a function of the velocity is shown in Fig 3B. It is noteworthy that the dependence is similar in both cases, for rate limiting ATP- (black curve) and ADP-binding (other curves). The finding is in agreement with the experimental observation that ADP affinity is the main mechanism of adaptation between fast and slow muscles [49]. Over a velocity range of 0.3 to 7 $\mu$m/s, the $K_D^{\mathrm{ADP}}$ scales approximately with $\sim v^2$ [51]. The predicted optimal ADP affinities are remarkably close to the quadratic law (dashed line in Fig 3B) over a wide interval of velocities.

## Anharmonic elastic potential

In the previous section, we saw that the stiffness of the elastic element had two conflicting effects. For motors at the beginning of the power stroke, a stiffer motor can produce more force. In a stiffer motor, transitions also take place in narrower intervals, further reducing the dissipation. On the other hand, motors that remain bound past $x = 0$ induce an effective drag. This raises the question whether efficiency can be improved by allowing an anharmonic spring. Asymmetric models for the myosin elasticity have been proposed previously in order to explain the measured properties of myosins [44, 52–54] or directly observed [55]. We have therefore relaxed the condition that the potential $U(x)$ be harmonic and allowed for an arbitrary shape, provided that $0 \leq U''(x) \leq K$. Numerically, we have divided the potential into a finite number of segments and run the optimization with the stiffness in each segment as a separate (constrained) parameter. The results show, however, that in all cases analyzed the optimal stiffness has a bimodal shape, reaching the constraint at $x > x_a$ and being zero ($U'' = 0$) for $x < x_a$, where $x_a$ is a transition point obtained by optimization.

The dashed lines in Fig 3A show the maximum efficiency if we allow anharmonic potentials. We observe an interesting transition: for slow velocities, a harmonic potential with the highest allowed stiffness is still optimal. However, at higher velocities, the optimal potential is asymmetric (Fig 3D). The shape of the potential reduces the negative forces caused by the post-powerstroke heads once they pass the potential minimum. At the same time, the remaining barrier, which is of the order of thermal energy, keeps the detached heads close to their unstrained position. In muscle myosin, such a potential can result from the elastic buckling of the tail domain and has indeed been observed in single molecule experiments [39, 56]. In finite ensembles of motors, asymmetric stiffness can have the consequence that the velocity becomes limited by the attachment, rather than the detachment rate [54, 57]. The softening of the spring in motors optimized for high velocities, on the other hand, is not seen in myosins. In fact, several comparisons between muscle types showed a higher stiffness in faster isoforms [58–61]. We note, however, that the measured stiffness is largely determined by the pulling motors ($x > 0$) and drawing conclusions on the regime with negative strain ($x < 0$) is difficult.

## Conclusions

We looked into the problem of myosin energetics from the reverse perspective: how would an optimally designed myosin motor work? We only consider a few physical constraints that cannot be arbitrarily altered in the course of optimization. These include the number of bound states, which is largely tied to the ATP hydrolysis cycle. We further constrain the maximum

stiffness, which we expect to be limited by the elastic properties of a protein. Finally, the second order rate constants of ATP and ADP binding are broadly conserved between myosins, which suggests that they are also close to their physical limits. Besides those constraints, our model makes some further assumptions. For one, we assume a single working cycle without any side branches, for example detachment in the ADP state. Such side branches generally take place between states with a higher free energy difference and therefore increase the dissipation (they can also be seen as leaks in the cycle). On the other hand, we neglect the discrete nature of actin subunits and assume that the myosin heads can bind anywhere on the actin filament. While the effect of discrete binding sites has been estimated as small [42], we expect that restricting the sites will somewhat increase the losses of the transition involving initial binding $(0 \to 1)$. The assumptions that the detachment rates are independent of force and that the other transitions depend in the simple way on the free energy difference present a restriction in the space of possible models. Furthermore, like virtually all myosin models, we assume that the stiffness of the elastic element is the same in all states. Also, we do not consider other optimization criteria that may be physiologically relevant, such as the maximum (isometric) force per motor or the ability of motors to work efficiently over a broader range of velocities. Beyond these constraints and assumptions, all parameters are the result of optimization. We identify three limiting factors for the efficiency: the number of bound states, the stiffness and the kinetics of ATP and ADP binding. Efficient motors require at least 3 bound states. The optimal free energy difference between the last two bound states (in myosin: ADP and without a nucleotide) depends strongly on the velocity for which the motor is optimized. Slow motors require an uphill transition, while fast ones work better with downhill. Another adaptation concerns the stiffness: at low velocities, a stiff potential allows them to exert a high force and to better control the strain dependent detachment rate. The importance of stiffness for a high efficiency was already highlighted in several muscle models [40, 41, 62, 63]. At high velocities the optimal potential becomes asymmetric: stiff for pulling forces and compliant in the pushing direction, above a certain threshold force. Many of these features are found in myosins: the chemical cycle has 3 distinct bound states [64], the largest power stroke takes place at the beginning of the cycle (connected to the release of Pi [65]), the energetics of the ADP release is the main source of variability between slow and fast myosins [51] and the elastic properties of the tail lead to an asymmetric potential [39]. A cross-comparison between different muscle types also shows a trade-off between speed and efficiency [51], as expected from our calculations. Similar observations were also made about a trade-off between power density and efficiency [66]— because the power output depends on more parameters than just the speed, it is more difficult to compare. Using concepts from stochastic thermodynamics, we were also able to identify the sources of dissipation inside the cycle, which can be more informative than just the efficiency of the cycle as a whole. We show that the largest free energy losses take place in the steps related to attachment and detachment from the actin filament. This shows that the rules governing the efficiency of non-processive motor proteins are very different from the processive ones like kinesin or F1-F0 ATP synthase, which were frequently studied from the perspective of stochastic thermodynamics [13, 22, 24, 25, 67, 68], and that the stochastic nature of attachment and detachment events leads to unavoidable losses.

## Supporting information

**S1 Fig. Stochastic simulation of a finite group of $N_m$ motors with a constant load.** Motors have $N_B = 2$ bound states and parameters obtained from the numerical optimization (Fig 2A in the main text). The upper row shows the simulated efficiency (symbols), compared with the stationary result (main text) for three different values of the dimensionless stiffness $\frac{1}{2} K d^2 / k_B T$.

The bottom row shows example traces (position $X$ vs. time $t$).
(PDF)

**S2 Fig. Stochastic simulation of a finite group of $N_m$ motors with $N_B$ = 3 bound states and optimized parameters.** The stationary solutions correspond to dotted lines in Fig 2A in the main text.
(PDF)

**S3 Fig. Stochastic simulation of a finite group of $N_m$ motors with $N_B$ = 3 bound states, two rate limiting steps and a harmonic potential and optimized parameters.** The stationary solution corresponds to solid lines in Fig 3A in the main text.
(PDF)

**S4 Fig. Stochastic simulation of a finite group of $N_m$ motors with $N_B$ = 3 bound states, two rate limiting steps and optimal potential (dashed lines in Fig 3A).**
(PDF)

**S5 Fig. Model in the limit of soft springs $Kd^2 \ll |\Delta G_{ATP}|$.** a) Potential landscape with $N_B$ = 2 bound states. b) The probability $P_2(t)$ to find the motor in state 2 at time $t$ after the attachment (black line). The red line shows the probability density of detachment at time $t$. c) Potential landscape with $N_B$ = 5 bound states. d) The probabilities $P_i(t)$, $i$ = 2, 3, 4, 5 to find the motor in state $i$ at time $t$ after attachment (dashed lines). The solid line shows the total probability $P_2(t) + P_3(t) + P_4(t) + P_5(t)$. The red line shows the detachment probability density. The larger number of states narrows the distribution of detaching motors and therefore their elastic energy before detachment.
(PDF)

**S1 Data. Values of optimized parameters corresponding to data in Figs 2A, 2B and 3A.**
(ZIP)

**S1 Source Files. Source code (C++) of the programs used to determine the optimal solutions (variable velocity, variable velocity outside hysteresis, fixed velocity).** The programs require the NAG Fortran library, Numerical Algorithms Group.
(ZIP)

## Author Contributions

**Conceptualization:** Andrej Vilfan.

**Data curation:** Andrej Vilfan, Andreja Šarlah.

**Formal analysis:** Andrej Vilfan.

**Investigation:** Andrej Vilfan, Andreja Šarlah.

**Methodology:** Andrej Vilfan, Andreja Šarlah.

**Software:** Andrej Vilfan, Andreja Šarlah.

**Validation:** Andrej Vilfan, Andreja Šarlah.

**Visualization:** Andrej Vilfan, Andreja Šarlah.

**Writing – original draft:** Andrej Vilfan.

**Writing – review & editing:** Andrej Vilfan, Andreja Šarlah.

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
