## [Decision Letter · Decision Letter 0]

19 Apr 2023

Dear Dr. Vilfan,

Thank you very much for submitting your manuscript "Theoretical efficiency limits and speed-efficiency trade-off in myosin motors" for consideration at PLOS Computational Biology. As with all papers reviewed by the journal, your manuscript was reviewed by members of the editorial board and by several independent reviewers. The reviewers appreciated the attention to an important topic. Based on the reviews, we are likely to accept this manuscript for publication, providing that you modify the manuscript according to the review recommendations.

Sincerely,

Stefan Klumpp

Academic Editor

PLOS Computational Biology

Jason Haugh

Section Editor

PLOS Computational Biology

Reviewer's Responses to Questions

**Comments to the Authors:**

Reviewer #1: In this manuscript, the authors theoretically study the design principles for the optimized dynamics of the non-processive molecular motor myosin. From well-established discrete state space descriptions, they derive analytical expressions for the efficiency and the entropy production associated with each transition between states. These quantities depend on the transition rates, free energies, and elastic-energy potentials. As their main result, they use a numerical method to find model parameters that maximize efficiency under different constraints. They identify two main design principles: i) slow motors perform optimally when the working stroke is driven by most of the free energy and their elasticity is very large; ii) fast motors need to spend most of the free energy at the last substep of ADP release and avoid interference forces by an asymmetric compliance.

Overall the manuscript presents new conceptual aspects to classify and better understand the general working principles of molecular motors. Before I can recommend publication, I would like the authors to improve their manuscript at several points:

1) Page 3, line 66: intuitively, I would have chosen alpha the other way around: detachment rates depend on the strain, but not the attachment rate. Maybe the authors could motivate their choice better.

2) Page 4, Fig. 1. caption: I would suggest a different terminology to distinguish between bound states and unbound states. The word “free” state sounds odd to me. “detached state” is also a possibility.

3) Page 5, Eq. 8, and Eq. 10, the authors should explain that the prime means a spatial derivative and the dot a temporal.

4) Page 6: Numerical solution: I understand that multi-parametric optimization is computationally difficult, but I would have liked to see a bit more details and discussion about the numerics. How are the initial guesses chosen? How sensitive are the results to the initial guesses? Are there any arguments why the optimized parameter set is not a local extrema, but the global one? Would it be possible to use a different optimization routine to check if the results are consistent?

5) Page 7, Table 1. As a suggestion for the authors, I am wondering if it would be possible to include more information in this table. Maybe by replacing “OPT” with the actual results from one optimization and then printing these numbers either italic or bold and explaining in the caption that these parameters were optimized. The advantage of such a representation would be to get an idea of the order of magnitude of the values without downloading and opening the zip files.

6) Page 8, Fig.2: (a) and (b) label of the axis: eta should be eta^max. Furthermore, the points c and d indicated in figure (a), and (b) could be potentially confusing. Please add a sentence in the caption about those points under (a) and (b), not at the end of the caption.

7) Page 9, line 194: “Although a slow transition leads to entropy production, it can improve the timing of the detachment and thereby reduce larger losses there.” Would it be possible to explain this sentence in more details? It is not clear to me what “improve the timing” means: longer attachment? shorter detached states? improved compared to what? I also don’t understand why an improvement in the timing reduces the losses.

Reviewer #2: In this manuscript, PCOMPBIOL-D-23-00402, Vilfan and Sarlah investigates the theoretical limit on energetic efficiency using models of the type proposed by Eisenberg and Hill with 2-4 attached mechano-chemical states, converging to 3 attached states as best, after initial tests. This is in good agreement with current consensus views from structural, biochemical and biophysical perspectives in the field (cf. Robert-Paganin et al., Chem Rev 2020; Hwang et al. PNAS 2021, Matusovsky et al., ACS Nano, 2021).

The authors attempt to determine the efficiency limits under different constraints to elucidate which properties of myosin that can be understood as adaptations to maximize efficiency. In their approach, they optimize all parameters by numerical efficiency optimization, excluding parameters bounded by physical/experimental limits e.g. the free energy of ATP, dimensionless stiffness and ADP-binding rate. The authors find that the efficiency depends on the number of states, the stiffness and the rate-limiting kinetic steps. Further, their result suggest a trade-off between speed and efficiency. In accordance with this general finding, their analysis predicts some interesting differences in the values of certain parameters between fast and slow motors.

The paper is well and clearly written and the rationale is well laid out in the Introduction, with generally adequate citation of the relevant literature.

The computational model is described in sufficient detail as is the implementation of the computations. The method to evaluate the origin of dissipation from entropy production is useful.

The studied subject is of interest and partly novel, although related types of general studies have been performed previously, as acknowledged by the authors in the Introduction. However, the present authors focus more strongly on non-processive motors such as the myosin II motors of muscle, rather than molecular motors in general or processive motors. I also find it of great value that the paper tends to bridge the gap between studies of molecular motors from a theoretical physical and chemical perspective on the one hand and biophysically/biochemically, largely experimentally, founded studies on the other. Often, otherwise, studies from these different perspectives seem to live their lives largely in parallel with limited interactions. This is often very clear by inspecting the reference list of respective papers. It is therefore refreshing to find several references from both groups of researchers in the current reference list. The use of the formalism of Hill is also well suited to the bridging activities as the formalism is fully comprehensible to most experimental biologists in addition to theoretical physicists and chemists (being created by a researchers that was mainly a theorist; TL Hill). The results from the optimizations allowing an anharmonic potential are interesting, giving consistently two segments with different stiffness, one less stiff at low cross-bridge strain and one stiffer at high strain.

On the basis of my above overall assessment, I view this paper favorably. However, there are some issues that need to be dealt with to improve the quality and to increase impact further as follows:

The authors discuss (in the Conclusions) possible limitations e.g. to what extent the results are generally valid, independent of the model used. However, I am not entirely convinced by the arguments. Could you please elaborate?

The authors also discuss the results in relation to other model studies with general aims, which they claim particularly concern processive motors. It would be great to know which papers the authors refer to in this case. Thus, please cite the relevant papers at this point as a quick look at some of the papers cited in the Introduction does not make it obvious that they do not at all apply to non-processive motors.

The lack of reproduction of the experimental efficiency vs velocity plot (Fig. 3; Line 204) seems problematic (if it is really the case?). For experiments see: (e.g. Barclay et al 2010 Prog Biophys Mol Biol). However, I might have misunderstood this; is the velocity here taken as maximum shortening velocity rather than the velocity variation for a given set of parameters vs load (in the force-velocity relationship)? Please clarify!

The results seem to suggest that slow motors work best if their stiffness is high whereas the opposite applies to fast motors (in addition to non-linearity of the stiffness). The predictions for the magnitude of stiffness seem to be in contrast with the real world data where optical tweezers based experiments (Capitanio et al., PNAS, 2006) have reported lower stiffness for slow than for fast myosin II motors of skeletal muscle.

I wonder if the authors could also extend their discussion to findings in a paper by Barclay (Clin Exp Pharmacol Physiol. 2017) reporting an inverse, linear relationship between maximum normalized power output and efficiency?

Minor

Lines 36-37. The definition of power density and force density may not be entirely clear to readers with a background in biology who might find interest in this paper. Please clarify.

Line 131. AFP->AFM. Please spell out “atomic force microscopy” upon first use.

Legend, Fig. 2. Which are the “thin lines” in panel b. I only find one thin line.

Line 178. Could be of value to exemplify “very soft springs” with illustration of the free energy diagrams for such a condition, e.g. in SI.

Lines 188-190. Based on the linearity in Fig. 2b, the limit of soft springs seems to apply for Kd2/2 up to almost 20 kBT. That does not seem to be consistent with Kd2<<delta-gatp .="" have="" i="" misunderstood="" something=""></delta-gatp>

**Have the authors made all data and (if applicable) computational code underlying the findings in their manuscript fully available?**

Reviewer #1: **No: **Simulation code and code for numerical optimization is not available at the moment

Reviewer #2: Yes

PLOS authors have the option to publish the peer review history of their article (what does this mean?). If published, this will include your full peer review and any attached files.

Reviewer #1: No

Reviewer #2: **Yes: **Alf Månsson

Figure Files:

Data Requirements:

Reproducibility:

References:

---

## [Editor Report · Decision Letter 1]

26 Jun 2023

Dear Dr. Vilfan,

We are pleased to inform you that your manuscript 'Theoretical efficiency limits and speed-efficiency trade-off in myosin motors' has been provisionally accepted for publication in PLOS Computational Biology.

Best regards,

Stefan Klumpp

Academic Editor

PLOS Computational Biology

Jason Haugh

Section Editor

PLOS Computational Biology

---

## [Editor Report · Acceptance letter]

18 Jul 2023

PCOMPBIOL-D-23-00402R1 

Theoretical efficiency limits and speed-efficiency trade-off in myosin motors

Dear Dr Vilfan,

I am pleased to inform you that your manuscript has been formally accepted for publication in PLOS Computational Biology. Your manuscript is now with our production department and you will be notified of the publication date in due course.

With kind regards,

Judit Kozma
